# Stabilizing Halogen-Bonded Complex between Metallic Anion and Iodide

**DOI:** 10.3390/molecules27228069

**Published:** 2022-11-21

**Authors:** Fei Ying, Xu Yuan, Xinxing Zhang, Jing Xie

**Affiliations:** 1Key Laboratory of Cluster Science of Ministry of Education, Beijing Key Laboratory of Photoelectronic/Electrophotonic Conversion Materials, School of Chemistry and Chemical Engineering, Beijing Institute of Technology, Beijing 100081, China; 2College of Chemistry, Key Laboratory of Advanced Energy Materials Chemistry (Ministry of Education), Renewable Energy Conversion and Storage Center (ReCAST), Tianjin Key Laboratory of Biosensing and Molecular Recognition, Shenzhen Research Institute, Frontiers Science Center for New Organic Matter, Nankai University, Tianjin 300071, China; 3Haihe Laboratory of Sustainable Chemical Transformations, Tianjin 300192, China

**Keywords:** halogen bond, metallic anion, nucleophilic substitution reaction, quantum chemistry calculation, reductive elimination

## Abstract

Halogen bonds (XBs) between metal anions and halides have seldom been reported because metal anions are reactive for XB donors. The pyramidal-shaped Mn(CO)_5_^−^ anion is a candidate metallic XB acceptor with a ligand-protected metal core that maintains the negative charge and an open site to accept XB donors. Herein, Mn(CO)_5_^−^ is prepared by electrospray ionization, and its reaction with CH_3_I in gas phase is studied using mass spectrometry and density functional theory (DFT) calculation. The product observed experimentally at m/z = 337 is assigned as [IMn(CO)_4_(OCCH_3_)]^−^, which is formed by successive nucleophilic substitution and reductive elimination, instead of the halogen-bonded complex (XC) CH_3_−I···Mn(CO)_5_^−^, because the I···Mn interaction is weak within XC and it could be a transient species. Inspiringly, DFT calculations predict that replacing CH_3_I with CF_3_I can strengthen the halogen bonding within the XC due to the electro-withdrawing ability of F. More importantly, in so doing, the nucleophilic substitution barrier can be raised significantly, ~30 kcal/mol, thus leaving the system trapping within the XC region. In brief, the combination of a passivating metal core and the introduction of an electro-withdrawing group to the halide can enable strong halogen bonding between metallic anion and iodide.

## 1. Introduction

The halogen bond (XB) is a type of non-covalent interaction that has attracted the interests of experimentalists and theoretical chemists in recent years [1,2,3,4,5,6,7,8,9]. According to the International Union of Pure and Applied Chemistry (IUPAC), “a halogen bond occurs when there is evidence of a net attractive interaction between an electrophilic region associated with a halogen atom in a molecular entity and a nucleophilic region in another, or the same, molecular entity” [10]. This definition states unambiguously that the halogen atom serves as an electrophile and interacts with a nucleophilic moiety. Typically, an XB is denoted as R−X···Y with three dots representing the bond, and X is a halogen atom (i.e., XB donor) that has an electrophilic region on its electrostatic potential surface, and Y is an XB acceptor. For the XB donor molecule (i.e., R−X molecule), the electrophilic (or positive) region on X, named as “σ-hole” [11,12], is induced by the R−X bond, which leaves an anisotropic distribution of electrons. The σ-hole magnitude, which represents the XB strength given by the same XB acceptor, scales with the polarizability of the halogen atom, that is, F < Cl < Br < I. Hence, changing the X atom can tune the XB’s strength, and there are other methods as well, including modifying the R-functional group and the electro-withdrawing ability of Y. The nature and tunability of XB make it useful in different fields spanning from material sciences to biomolecular recognition and drug design [13,14,15,16,17,18]. 

The common XB acceptors are nucleophiles, such as N, O, S, P, or halogen atoms/anions; metal anions are rarely seen. This is because atomic metal anions are usually too reactive towards organohalogens. For example, the reaction between Au^−^/Ag^−^/Cu^−^ anions and CH_3_I in gas phase give rise to a Grignard reagent-like product [CH_3_−M−I]^−^, where a covalent M−I bond is formed [19,20]. This structure is calculated to be ~2.0–3.0 eV more stable than the XB complex [CH_3_−I···M]^−^ [20,21]. To achieve the goal of forming metallic acceptor-containing halogen bonds, one of our authors proposed two strategies: one is to utilize a metal cluster anion with a high electron detachment energy; the other is to design a ligand-passivated/protected metal core that can maintain the negative charge [22]. The goal of this work is to check the feasibility of the second strategy experimentally. Hence, herein, we prepared a Mn(CO)_5_^−^ anionic compound by electrospray ionization and investigated its reactivity with CH_3_I.

To test whether Mn(CO)_5_^−^ anion is a suitable candidate to form a halogen-bonded complex, we first investigated the properties of Mn(CO)_5_^−^ anion by density functional theory (DFT) calculation using M06-2X method [23] with aug-cc-pVTZ basis set [24,25,26]. As shown in Figure 1a, the structure of Mn(CO)_5_^−^ anion has a pyramidal shape, with one CO ligand in the horizontal direction and the other four CO ligands almost in the same plane (see Appendix A for an illustration), leaving the left an open site to accept a XB donor. A Mulliken charge analysis [27] (Figure 1a) indicated that the Mn-atom core is the most negative, with a charge of −0.84 e, and this is clearly displayed in the electrostatic potential map (Figure 1b). In addition, the HOMO of Mn(CO)_5_^−^ anion (Figure 1c) comprising C p orbital and Mn $d_{x^{2}}$ orbital has electrons evenly delocalized on four C atoms, thus stabilizing the compound. In brief, Mn(CO)_5_^−^ anion fulfills the two criteria of the second strategy: the metal core is negatively charged and has at least one open site to accept the XB.

In this work, we will first study the reaction between Mn(CO)_5_^−^ anion and CH_3_I in gas phase using a linear ion trap mass spectrometer. Then the products and mechanism will be analyzed with the help of DFT calculation. The stability of the halogen-bonded complex is evaluated, and a strategy is proposed to further stabilize it.

## 2. Methods

### 2.1. Experimental Methods

Mass spectra were acquired using a linear ion trap mass spectrometer (LTQ-XL, Thermo-Fisher, Waltham, MA, USA). The inlet capillary temperature of the mass spectrometer was maintained at 275 °C. The tube lens voltage on the LTQ-XL was set to be 0 V in order to avoid in-source fragmentation of the fragile species. The applied negative voltage was set at −4000 V in this study in order to trigger the electrospray ionization. A methanol solution of Mn_2_(CO)_10_ was sprayed to generate the Mn(CO)_5_^−^ anion. The collision-induced dissociation (CID) spectrum of the Mn(CO)_5_^−^ anion is presented in Appendix A. Gas-phase reaction between the Mn(CO)_5_^−^ anion and the neutral CH_3_I molecule at room temperature was conducted in the linear ion trap by using the collision-induced dissociation (CID) mode that is, the MS^2^ mode of the mass spectrometer in order to isolate the Mn(CO)_5_^−^ anion in the trap. The CH_3_I molecules were introduced to the trap by putting a drop of CH_3_I into a small stainless-steel reservoir that was connected to the pipeline of the He collision gas. The collision energy was set to be under 10 V in order to trigger the reactions. 

### 2.2. Computational Methods

Geometry optimizations were performed using M06-2X functional [23], with aug-cc-pVTZ basis set [24,25,26] used for H, C, O, F, and Mn atoms, and aug-cc-pVTZ-PP basis set [28,29] used for I atoms. Various configurations were optimized for CH_3_I−Mn(CO)_3_^−^, and the most stable structures were used for discussion (see Appendix A for details). Harmonic vibrational frequencies were calculated to confirm the nature of the stationary points. Intrinsic reaction coordinate (IRC) calculations were performed on transition states to confirm that they connected the correct intermediates. The ground state of Mn(CO)_4_I^−^ is doublet, and the other metal-involved species are all singlet. The zero-point corrected energy is used in the potential energy profile. Gaussian 16 [30] package was used to perform all the calculations.

## 3. Results and Discussion

### 3.1. Mass Spectrometry

A typical mass spectrum showing the reaction products between Mn(CO)_5_^−^ and CH_3_I is presented in Figure 2a. Three major peaks at m/z 281, 294, and 337 were observed, corresponding to the masses of CH_3_I-Mn(CO)_3_^−^, Mn(CO)_4_I^−^, and CH_3_I-Mn(CO)_5_^−^, among which the latter is the direct product from the reaction between Mn(CO)_5_^−^ and CH_3_I, but the former two are the collision fragments of the latter. To obtain structural information for the m/z 337 peak, we further isolated it with the MS^3^ mode of the mass spectrometer; its CID fragments with a collision energy of 5 V are presented in Figure 2b. If CH_3_I-Mn(CO)_5_^−^ is a weakly bonded species of Mn(CO)_5_^−^ and CH_3_I, its fragments should predominantly be Mn(CO)_5_^−^. However, two distinct fragments, CH_3_I-Mn(CO)_4_^−^ and MnIO_2_^−^, were observed, suggesting that the m/z 337 peak is not a weakly bound species.

### 3.2. Density Functional Theory Calculation

#### 3.2.1. Mn(CO)_5_^−^ + CH_3_I Reaction Mechanism

To identify the structure and understand the formation mechanism of the aforementioned observed products, we performed DFT calculations. Figure 1 depicts the potential energy surfaces (PESs) for Mn(CO)_5_^−^ + CH_3_I, and selected structures are displayed in Figure 3. Enthalpy and free energy values at 298.15 K are listed in Table 1.

Figure 3 depicts that a halogen-bonded complex (XC) CH_3_−I···Mn(CO)_5_^−^ is formed by an I atom attacking the open site of Mn, and XC is 5.3 kcal/mol lower in energy than the reactants. The I···Mn distance within XC is 3.652 Å, which is 79% of the sum of the van der Walls radii of I (2.36 Å) and Mn (2.24 Å). At the same time, a slightly more stable pre-reaction complex (RC) is formed between a C atom interacting with Mn; it is lower in energy by 2.8 kcal/mol. Of note, additional conformers of RC that are higher in energy are localized: one has a linear I−C−Mn shape, and the other I−C−Mn angle is ~ 90°. These two structures are structural isomers of RC, which may appear when CH_3_I attacks Mn(CO_5_)^−^ in a different direction. For clarity, they are omitted in Figure 3 and are instead present in Appendix A. After crossing a back-side attack nucleophilic substitution barrier (TS1) of 8.3 kcal/mol, it proceeds to post-reaction complex PC1. We also considered the front-side attack S_N_2 transition state (TS2); however, it is too high (34.4 kcal/mol) to occur. Within PC1, CH_3_ fragment and I fragment are located on the opposite side of Mn with a weak I-C interaction. Relative to the reactants, PC1 is −32.0 kcal/mol in energy. Then PC1 can undergo a reductive elimination barrier (TS3) of 11.8 kcal/mol and ends up with the formation of a C−C bond and the migration of I to bond with Mn. This resulted complex (PC2) is very stable, and is −43.2 kcal/mol relative to the reactants. Because TS1 is almost thermally neutral and TS3 is lower in energy than the reactants, the most stable PC2 can be formed under room temperature (the experimental condition). For this reason, in Figure 2, the signal at 337 m/z was assigned to be PC2, and it agrees with experimental results that this species is quite stable. 

The calculated energy for Mn(CO)_5_^−^ + CH_3_I → Mn(CO)_4_I^−^ + COCH_3_ reaction is −5.0 kcal/mol, and the calculated energy to form Mn(CO)_4_I^−^ + CO + CH_3_ is 5.7 kcal/mol. Therefore, we believe the experimentally observed Mn(CO)_4_I^−^ is more likely to be dissociated from PC2 and to generate COCH_3_ at the same time, and is less likely to be caused by the collision-induced dissociation that forms CO and CH_3_. 

The calculated energy from generating CH_3_I−Mn(CO)_3_^−^ + 2CO from reactants is −6.1 kcal/mol downhill. The most stable structure of CH_3_I−Mn(CO)_3_^−^ has a pseudo-bipyramidal shape (Figure 3). Analyzing the PES indicates that it may dissociate from PC1 or, provided sufficient energy, dissociate from TS2. 

In brief, although there is considerable halogen bonding between CH_3_I and Mn(CO)_5_^−^, the passivated Mn center within Mn(CO)_5_^−^ is still reactive towards CH_3_I, thus making the XC a transient species. The observed CH_3_I-Mn(CO)_5_^−^ signal is PC2, which forms by nucleophilic substitution and the following reductive elimination. 

#### 3.2.2. Stabilizing Halogen-Bonded Complex by CF_3_I 

It is known that introducing an electron-withdrawing group, such as F, to the methyl group can increase the σ-hole magnitude, and thus the XB strength. Therefore, we changed CH_3_I to CF_3_I, which induces a greater positive region on the I atom, in the hope that it can stabilize the halogen-bonded complex when interacting with Mn(CO)_5_^−^. On the other hand, changing CH_3_ to a heavier CF_3_ group is expected to raise the inversion S_N_2 barrier [31], thus preventing the S_N_2 reaction. This may also help trap the system in a halogen-bonded complex well, so we computed the PES of Mn(CO)_5_^−^ + CF_3_I. 

As shown in Figure 1, the halogen-bonded complex (XC’) CF_3_−I···Mn(CO)_5_^−^ well is 17.5 kcal/mol deep, where the pre-reaction complex RC’ is 14.3 kcal/mol higher than it is. Within XC’, the I···Mn distance is 3.224 Å, being 0.428 Å shorter than the corresponding value of XC. This is consistent with XC’ being more stable than XC. To characterize the interaction between CR_3_I and Mn(CO_5_)^−^ within the halogen-bonded complex CR_3_−I···Mn(CO)_5_^−^, natural bond orbital (NBO) [32,33] calculations were performed for R = H and F in order to analyze the donor–acceptor charge transfer properties. As shown in Figure 4, taking CF_3_−I···Mn(CO)_5_^−^ as an example, the donor orbital is the Mn−C bonding σ orbital and Mn 3p orbital, and the acceptor orbital is the C–I antibonding σ* orbital. The same scenario also applies for CH_3_−I···Mn(CO)_5_^−^. In comparison, when the halogen-bonded complex is composed of a main group nucleophile, such as F^−^ and CH_3_I (i.e., [CH_3_−I···F]^−^), the donor NBO is a 2p orbital; when the nucleophile is Cu^−^/Ag^−^/Au^−^, the donor NBO is an s orbital [21,34]. 

Additionally, the back-side attack S_N_2 barrier (TS1′) is largely raised to 28.9 kcal/mol relative to the reactants. Although the front-side attack transition state (TS2′) is lower than TS2, TS2′ is still 26.1 kcal/mol uphill. Of note, different from CH_3_I, the front-side attack S_N_2 barrier (TS2′) is lower than the back-side attack S_N_2 barrier (TS1′) by 2.8 kcal/mol. If the reactants are cooled to room temperature or even lower, they are unlikely to cross these barriers or proceed to nucleophilic substitution and the following reductive elimination. Of note, the PC1′ and PC2′ complexes are even lower than PC1 and PC2, but PC1′ needs to cross a barrier (TS3′) of 37.7 kcal/mol. In another words, if the system crosses TS1′, both PC1′ and PC2′ can be formed and stable.

To summarize, calculations show that replacing CH_3_I with CF_3_I stabilizes the halogen-bonded complex and raises the nucleophilic substitution barrier. This is an effective strategy to obtain strong halogen bonding between iodide and metallic anions.

## 4. Conclusions

To achieve the goal of constructing a stable halogen-bonded complex between metallic anionic species and halide, we adopted the strategy of passivating the reactive metallic anion by introducing protected ligands. Thus, we designed the Mn(CO)_5_^−^ anionic compound, and DFT calculation confirms that it maintains a negatively charged core and has an open site to accept halogen bond donors. Next, the Mn(CO)_5_^−^ species was prepared by electrospray ionization and then reacted with CH_3_I in gas phase using a linear ion trap mass spectrometer. The major products were CH_3_I-Mn(CO)_3_^−^, Mn(CO)_4_I^−^, and CH_3_I-Mn(CO)_5_^−^. DFT calculations suggested that CH_3_I-Mn(CO)_5_^−^ is a stable species (i.e., [IMn(CO)_4_(OCCH_3_)]^−^) that forms by nucleophilic substitution and reductive elimination. The halogen-bonded complex CH_3_−I···Mn(CO)_5_^−^ could be a transient species because the interaction between I and Mn is weak. 

By substituting CH_3_I to CF_3_I, calculations predicted that the resulted halogen-bonded complex CF_3_−I···Mn(CO)_5_^−^ is stabilized considerably. In addition, the barrier for nucleophilic substitution was greatly raised, allowing the system to trap in the XB complex well, given that the system is cool enough to avoid crossing the S_N_2 barrier. This work presents an example of stabilizing the halogen bonding between a ligand-protected metal anion and halide with strong electro-withdrawing group. By adopting a similar strategy, it is anticipated that more metallic acceptor-containing XBs will be discovered. 

## Data Availability

Data is contained within the article or Appendix A.

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
