# Peer review of "Stabilizing Halogen-Bonded Complex between Metallic Anion and Iodide"

_molecules, 2022, doi:10.3390/molecules27228069_

Round 1

Reviewer 3 Report

In this manuscript, Authors presented a way to construct a stable halogen-bonded complex between metallic anionic species and halide. The strategy of passivating the reactive metallic anion by introducing protected ligands has been accepted. The pyramidal-shaped [Mn(CO)5]− anion is suggested as a potential metallic halogen bond (XB) acceptor with a ligand-protected metal core which maintains the negative charge and an open site to accept XB donor.

Mn(CO)5]− was prepared by electrospray ionization and its gas-phase reaction with CH3I was studied using mass-spectrometry and Kohn-Sham method calculations. Geometry optimizations were performed with M06-2X functional, with aug-cc-93 pVTZ basis set for H, C, O, F, Mn atoms, and aug-cc-pVTZ-PP basis set for I atom. Intrinsic reaction coordinate (IRC) calculations, performed on transition states, confirmed that  the proper intermediates are connected. The experimentally observed moiety, [IMn(CO)4(OCCH3)]−, was formed by successive nucleophilic substitution and reductive elimination, instead of the halogen-bonded complex CH3−I···Mn(CO)5−.

Kohn-Sham calculations show that replacing CH3I to CF3I strengths the halogen bonding within the halogen-bonded complex due to the electron-withdrawing ability of F atom. At that, the nucleophilic substitution barrier raises significantly (up ~ 30 24 kcal/mol); that leaves the system trapping within a complex.  Authors conclude that the combination of passivating metal core and electro-withdrawing group to halide enables strong halogen bonding between metallic anion and iodide.

           This is careful work which can be published. Nevertheless, some improvements might be done.

1) Presence of the Mn atom demands the relativistic calculations. Otherwise, the sigma-hole can be erroneously "identified".

2) Sigma-hole concept is valid at the large distances between interacting pieces. When the distance is small, complex does exist as the unified entity. It should be commented in the text.   
